# Optimization based data enrichment using stochastic dynamical system models

**Griffin M. Kearney** [1,2]*, **Makan Fardad**[2]

**1** OpB Data Insights LLC, Syracuse, NY, United States of America, **2** Department of Electrical Engineering and Computer Science, Syracuse University, Syracuse, NY, United States of America

* griffin.kearney@opbdatainsights.com

**Data Availability Statement:** The data has been published in the following repository: Kearney,

## Abstract

We develop a general framework for state estimation in systems modeled with noise-polluted continuous time dynamics and discrete time noisy measurements. Our approach is based on maximum likelihood estimation and employs the calculus of variations to derive optimality conditions for continuous time functions. We make no prior assumptions on the form of the mapping from measurements to state-estimate or on the distributions of the noise terms, making the framework more general than Kalman filtering/smoothing where this mapping is assumed to be linear and the noises Gaussian. The optimal solution that arises is interpreted as a continuous time spline, the structure and temporal dependency of which is determined by the system dynamics and the distributions of the process and measurement noise. Similar to Kalman smoothing, the optimal spline yields increased data accuracy at instants when measurements are taken, in addition to providing continuous time estimates outside the measurement instances. We demonstrate the utility and generality of our approach via illustrative examples that render both linear and nonlinear data filters depending on the particular system. Application of the proposed approach to a Monte Carlo simulation exhibits significant performance improvement in comparison to a common existing method.

## Introduction

In most practical applications, measurements are noisy observables sampled from underlying dynamics that are governed by physical laws and polluted by stochastic processes. Knowledge of both the governing temporal dynamics and the distribution of the uncertainty can be exploited to add accuracy to the measurements and counter the effect of noise. It also allows for interpolation and extrapolation, to obtain estimates for those time instances at which no measurements were made. Here we collectively refer to data accuracy improvements and completion as data enrichment.

In this work, a novel technique for the enrichment of data governed by stochastic dynamics and observed through noisy measurement systems is developed. We adopt the term data enrichment to describe the mitigation of errors in measurements (I.E. the reduction of noise) and the estimation of missing data points (I.E. interpolation / extrapolation of measurements)

Griffin (2024), ""Optimization Based Data Enrichment Using Stochastic Dynamical System Models" Monte Carlo Dataset", Mendeley Data, V1, doi: 10.17632/ygx3fbr7rc.1.

**Funding:** Makan Fardad: CAREER CMMI-1750531, National Science Foundation, https://www.nsf.gov/ This funder played no role in the study design, data collection and analysis, decision to publish, or preparation of the manuscript.

**Competing interests:** The authors have declared that no competing interests exist.

simultaneously. For instance, if the position of a vehicle were periodically measured in time using a noisy GPS system, then data enrichment would improve the accuracy of the GPS measurements at the sampling times and provide vehicle position estimates for the times in which no GPS samples have been taken.

Our new approach achieves this goal through joint analysis of stochastic dynamical systems and stochastic measurements. Data accuracy improvements and completion are achieved by generating maximum likelihood estimates using the distributions that characterize the stochastic dynamics and measurements. In the present work, since our attention is on dynamical systems, our method results in optimal splines over time.

A spline refers to a piecewise function with a prescribed structure on each of the underlying sub-intervals that support it over the domain. Spline techniques are well-established and among them cubic spline interpolation is used across a variety of applications [1–3]. However, the classical construction of splines often requires subjective assumptions to be enforced in the fitting process which lack physical models. In our treatment this is not the case. The spline shape and the constraints enforced to optimally enrich the data are determined entirely by the stochastic dynamics and measurements within the maximum likelihood estimation framework. In other words, our technique automatically provides optimally structured splines for each system of interest.

The way in which the new optimization framework renders splines eliminates the need to specify empirical constraints providing a notable advantage when compared to classical spline constructions. The "natural" cubic spline (NCS) is a classic spline type that continues to be used in current research [3]. In this technique piece-wise cubic functions are stitched together at data points. It is commonly required that the spline coincide with measured data at measurement times, that the spline is $C^2$ continuous (i.e., it is a continuous function with continuous first and second derivatives), and that the second derivative is forced to vanish at the first and final data points. Using our framework we demonstrate that cubic splines are indeed the optimal spline shape for a simple, common class of stochastic dynamics, thus providing new novel insights that further justify their use. However, rigid fitting constraints are avoided in our proposed method which renders larger classes of splines that are less sensitive to measurement noise in comparison to classic methods such as NCS.

B-splines are an increasingly popular alternative which also offer noise sensitivity benefits in comparison to classic treatments. The methods used to construct B-splines enable mitigation of measurement noise, like in our proposed framework. Gesemann provides a nice overview of how these splines are constructed in fluid dynamic applications [4]. In this method, rendering the B-spline to mitigate measurement noise requires tuning of data filter parameters on an ad hoc basis after examination of noise power densities which requires substantial effort. By contrast, our framework generates adaptive fitting conditions that apply to general measurement and process noise without ad hoc tuning requirements.

Kalman smoothing is an alternative well-established method for data enrichment that does not generate spline outputs. There are similarities in the reasoning behind our framework and the widely used Kalman approach described in his two seminal papers [5, 6]. This powerful technique was popularized for real-time online data enrichment in aerospace navigation, but continues to perform well and warrant additional study in current research [7–10]. Kalman methods provide estimates by balancing dynamic and measurement stochastic process likelihoods simultaneously. Similar to Kalman smoothing, we assume knowledge of underlying system dynamics which is incorporated to adaptively match the filter to a data system through optimization. Despite this high-level similarity, the present optimization method and the Kalman method are fundamentally different. Kalman assumes, from the outset, that the filter acts linearly on observed measurements whereas our method requires no such restriction. This

limitation results in sub-optimal filter performance outside of linear dynamical systems under Gaussian processes [11]. Indeed, the optimality of the linear filter follows naturally from our framework when considering these systems without such an assumption; it is an automatic consequence of our technique. Moreover, classical implementation of Kalman smoothing requires solving (non-linear matrix-valued) Riccati equations which are computationally prohibitive in large problems. In our construction we only require the solution of *linear* equations when considering Kalman-optimal (linear dynamics, Gaussian processes) data systems which simplifies implementation.

Gaussian process regression (GPR) is another related and noteworthy technique for enriching many diverse data systems, and overviews of these methods are becoming more available [12–14]. GPR has experienced increased popularity and been the focus of research recently due to appealing applications in artificial intelligence and machine learning. Similar to the goals of this work, GPR methods seek to fit functions to sets of discrete, noisy data measurements using probability maximization. Informative tutorials on GPR are presented in [12, 14], and these works demonstrate how the methods rely heavily on underlying model kernels which must be specified by a user. Developing GPR kernels within implementations can be informed using underlying physics, but challenges still arise even when a priori knowledge is available. Indeed, the work of [13] focuses on "physics-informed machine learning" and provides a good survey of the obstacles encountered in constrained GPR. Our work is built on rich dynamical systems theory and stochastic processes, and as a result we incorporate physics through rigorous first-principles that aid us greatly in circumventing these challenges. Finally, as the name implies, GPR requires that the underlying data be modeled by a *Gaussian* stochastic process. While we include many Gaussian process problems for illustrative examples, we are not limited to these and we also demonstrate application of our proposed method on non-Gaussian systems.

To summarize, classic splines are simple but sensitive to errors due to rigid constraints or require substantial ad hoc tuning. Kalman smoothing is robust, well-established, and informed by foundational physics, but the implementation suffers from inefficiencies due to their original design purpose for real-time aerospace vehicle state estimation. GPRs produce rich outputs that interact well with measurement and process noise, but challenges arise in the incorporation of underlying physics and are not generally applicable to non-Gaussian systems. Our proposed method generates flexible, adaptive splines that respect underlying system dynamics and stochastic processes gracefully. We are able to do this for a relatively large class of problems, including those with non-linear dynamics and non-Gaussian processes. However, when the dynamics are linear and processes Gaussian then our framework simply requires the solution of an elegant linear system which enables one to use many efficient techniques that are practical even for large data systems.

We proceed with development of the new method by first describing required preliminaries in the Model Overview and Preliminaries section. During the preliminary discussion we motivate and establish a critical extension of instantaneous process distributions to continuous intervals which is required for forming the general problem. In Motivational Example section we provide a simple example to demonstrate how the problem is formed by considering the motion of a point mass, but refrain from computing the solution at this stage. The example is provided to make the general treatment of the problem in more approachable in the General Problem and Analysis section, and it is within this discussion that we develop the optimization conditions in detail. Finally, in the Illustrative section we examine fundamental examples that demonstrate how optimal splines are constructed using the main theorem that results from the general analysis with simulated data.

A summary of the contributions of this work are as follows:

1. We develop a novel optimization framework to generate continuous time enrichments of discretely sampled systems with known stochastic dynamical process and measurement model distributions. This development includes a new method for extending instantaneous process distributions to continuous intervals.

2. We establish the necessary optimization criteria which characterize solutions of the new framework. The optimization criteria are provided as a set of general ordinary differential equations which determine the optimal spline structure, and a set of algebraic equations which characterize the boundary conditions at measurement points. These conditions are derived through application of the Calculus of Variations.

3. When the governing dynamics are linear, and both the dynamical process and measurements are subject to Gaussian stochastic processes, then we show that the new theory results in optimal splines that behave as linear filters on noisy measurements. We demonstrate that this produces cubic splines in a simple illustrative set of dynamics (simple particle motion), but that it also produces non-cubic splines in a second illustrative case (simple harmonic motion). We then analyze an example on the general analysis of the optimal splines for Kalman systems (linear dynamics with Gaussian stochastic distributions) which concretely demonstrates that the linear optimal solution flows naturally from the new theory without prior assumption. Finally, to demonstrate generality, we conclude with consideration of two examples that produce non-linear optimal solutions. The first of these examines an example of non-Gaussian process noise, and the second involves non-linear governing dynamics.

## Model overview and preliminaries

The variable $x$ is used to represent the state of a system, and the variable $y$ is used to represent measurements observed within it. Throughout this work when a vector-valued function of time, say $x$, appears without explicit reference to $t$ then it implies the function over its entire temporal support, i.e. $x(\cdot)$. We begin by assuming a general form of governing dynamics on the state $x$,

$$\dot{x}(t) = f(t, x(t)) + v(t, v(t)), \tag{1}$$

and a general form of measurements

$$y(t) = g(t, \dot{x}(t)) + h(t, x(t)) + \xi(t, w(t)). \tag{2}$$

For each $t$ the vector $x(t)$ is assumed to be of dimension $n_x$ and the vector $y(t)$ of dimension $n_y$. The functions $f$, $g$, $h$, $v$, and $\xi$ are permitted to be arbitrary, emphasizing that in this framework we allow for non-linear dynamics and measurements. Moreover, these functions are also allowed to have explicit dependence on time.

In general, $v$ and $w$ are used to denote stochastic processes, they are vectors of dimensions $n_v$ and $n_w$ respectively, and we assume them to be statistically independent of one another. At each instant $t$, $v(t)$ and $w(t)$ denote random variables, and both are associated with corresponding probability density functions $\rho_v(t, v(t))$ and $\rho_w(t, w(t))$, which characterize their distributions. Furthermore we assume that the images of $v$ and $w$ are statistically independent of themselves at distinct time instants. In other words, $v(t_0)$ is independent of $v(t_1)$ for $t_1 \neq t_0$.

The dynamical stochastic process (1) is assumed to hold for all times $t$. The stochastic measurement description (2) is only defined for a strict and possibly continuous subset of time

instants. We denote the entire time horizon under consideration as $\mathscr{T}$, and the subset where measurements have been captured as $\mathscr{T}_M \subseteq \mathscr{T}$.

In many practical problems $\mathscr{T}_M$ will be a finite collection of disjoint discrete points, but continuous measurements on intervals are also permissible. Our goal will be to quantify the likelihood of outcomes on the set $\mathscr{T}$, and for the points in $\mathscr{T}_M$ this is determined by both the dynamical process and measurement randomness as modeled through $v$ and $w$. On the other hand, for the set $\mathscr{T} - \mathscr{T}_M$ the likelihood is only determined by $v$ as no measurements are taken at these times. We have assumed that the dynamics and measurement stochastic processes are independent, which allows us to write the piecewise function

$$\rho(t) = \begin{cases} \rho_v(t, v(t)) & t \in \mathscr{T} - \mathscr{T}_M \\ \rho_v(t, v(t))\rho_w(t, w(t)) & t \in \mathscr{T}_M, \end{cases} \tag{3}$$

for any time $t \in \mathscr{T}$. The function $\rho$ assigns a real number, the probability density, to the value $v(t)$ (or to the pair $v(t)$, $w(t)$ when measurements are present) at the time $t$. It will be important to keep in mind, once we form the maximum likelihood problem, that $\rho$ is a function of $v$ and $w$ and depends on their probability distributions even though these dependencies are not made explicit in our choice of notation.

Computing probabilities at discrete points is possible using this function directly. Suppose that $t_0$ and $t_1$ are two time instants of interest at which no measurements are taken. We assume that the process values $v(t)$ are independent for distinct times, and therefore the joint distribution of $v(t_0)$ and $v(t_1)$ is expressed as

$$\rho(\{t_0, t_1\}) = \rho(t_0)\rho(t_1) \tag{4}$$

$$= \rho_v(t_0, v(t_0))\rho_v(t_1, v(t_1)), \tag{5}$$

a simple product. This can be done for a set of more than two discrete points by including more factors in the product, and adjusted using (3) for times when measurements are taken.

Difficulties arise applying this approach when we must compute the probability over intervals instead of discrete points. Assume that $\tau = (t_0, t_1)$ is an open interval over which no measurements are taken. It is tempting to write

$$\rho(\tau) = \prod_{t \in (t_0, t_1)} \rho_v(t, v(t)), \tag{6}$$

but it is unclear how the product in this expression should be computed or if it is even well-defined as it consists of an uncountably infinite number of factors.

We resolve this difficulty by proposing a method for extending instantaneous distributions in the S1 Appendix. The treatment provides analytic machinery for extending functions described by $\rho$ to functionals on continuous intervals that we denote with $\mu(\tau, \rho)$. The extension is of the form

$$\mu(\tau, \rho) = e^{\frac{1}{|\tau|}\int_\tau \ln \rho(t)\ dt}, \tag{7}$$

where $|\tau|$ denotes the length of the interval, and as a point of convention if $\tau = \{t\}$, then we define

$$\mu(\tau, \rho) = \rho(t) \tag{8}$$

for simplicity of notation.

The extension $\mu(\tau, \rho)$ assigns a real number to the image of the process $v$ (or to the pair of images $v$, $w$ on intervals with measurements) on $\tau$. We interpret $\mu$ as an analog of the probability density, but on interval/process pairs instead of discrete time/random variable pairs. In alignment with this interpretation, we define how the joint likelihood over two separate intervals is treated to maintain the parallel. We say that two intervals $\tau_0$ and $\tau_1$ are separated if there exists some point between them. For instance $(a, b)$ and $(b, c)$ are separated for all $a < b < c$ since the point $\{b\}$ lies between them, but $(a, b]$ and $(b, c)$ are not separated. If $\tau_0$ and $\tau_1$ are separated intervals, then we require the extension function satisfies the relation

$$\mu(\tau_0 \cup \tau_1, \rho) = \mu(\tau_0, \rho)\mu(\tau_1, \rho). \tag{9}$$

In all problems considered here, the sets $\mathscr{T}_M$ and $\mathscr{T} - \mathscr{T}_M$ will permit representations as unions of separated intervals and discrete points. The maximum likelihood problem we propose has an objective of the general form

$$J = \mu(\mathscr{T} - \mathscr{T}_M, \rho)\mu(\mathscr{T}_M, \rho), \tag{10}$$

which is maximized subject to the governing dynamics (1)–(2). Eq (9) is used to simplify each of the two terms on the right-hand-side of (10) for a given problem. We recall from (3) that $\rho$ depends on $v$, $w$ and their probability distributions. Loosely speaking, the maximum likelihood problem seeks to find the most likely trajectories for $v$ and $w$ that satisfy the constraints (1)–(2).

In the General Problem and Analysis section we fully develop and analyze the general problem. Before we do this however, we use the next section to formulate the optimization problem on an illustrative example to motivate the general treatment.

## Motivational example

We focus attention on a simple scenario. We consider the one dimensional undamped motion of a point mass under Gaussian stochastic forcing. Position measurements are collected periodically in time during the particle's motion, which are subject to Gaussian errors.

The governing dynamics of the system are

$$\ddot{r}(t) = v(t) \tag{11}$$

where $r(t)$ is a real number representing the position at time $t$ and $v$ is a scalar stochastic process with

$$\rho_v(t, v(t)) = \frac{1}{\sqrt{2\pi}\sigma_p} e^{-\frac{v(t)^2}{2\sigma_p^2}}. \tag{12}$$

We denote the process noise variance as $\sigma_p^2$ in (12) and assume that it is known. The position measurements are modeled

$$y(t) = r(t) + w(t) \tag{13}$$

where $y(t)$ denotes the noisy measurement at time $t$ and $w$ is a stochastic process with

$$\rho_w(t, w(t)) = \frac{1}{\sqrt{2\pi}\sigma_m} e^{-\frac{w(t)^2}{2\sigma_m^2}}. \tag{14}$$

We denote the measurement noise variance as $\sigma_m^2$ and also assume that it is known. The state space vector is written as

$$x(t) = \begin{pmatrix} r(t) \\ \dot{r}(t) \end{pmatrix} \tag{15}$$

and enables us to express this system in standard form:

$$\dot{x}(t) = \begin{pmatrix} 0 & 1 \\ 0 & 0 \end{pmatrix} x(t) + \begin{pmatrix} 0 \\ 1 \end{pmatrix} v(t) \tag{16}$$

$$y(t) = \begin{pmatrix} 1 & 0 \end{pmatrix} x(t) + w(t). \tag{17}$$

Suppose that measurements are taken periodically in time with sampling frequency $f_0$. We represent the sampling times with the set $\{t_k\}_{k=0}^K$ and assume

$$t_{k+1} - t_k = \frac{1}{f_0} \tag{18}$$

for $k = 0, 1, \ldots, K - 1$. Let us consider the time interval $\mathscr{T} = [t_0, t_K]$, where $t_0$ is the first and $t_K$ is the final measurement time. We have

$$\mathscr{T}_M = \{t_k\}_{k=0}^K, \tag{19}$$

by assumption, and

$$\mathscr{T} - \mathscr{T}_M = \cup_{k=0}^{K-1}(t_k, t_{k+1}) \tag{20}$$

as a result. The set $\mathscr{T}_M$ consists of separated discrete points, and $\mathscr{T} - \mathscr{T}_M$ consists of separated continuous intervals. This allows us to use relation (9) to form the objective (10) for this example. Namely,

$$\mu(\mathscr{T} - \mathscr{T}_M, \rho) = \prod_{k=0}^{K-1} \mu((t_k, t_{k+1}), \rho) \tag{21}$$

and

$$\mu(\mathscr{T}_M, \rho) = \prod_{k=0}^{K} \mu(\{t_k\}, \rho). \tag{22}$$

Eqs (12) and (14) are used in (3) to define $\rho$ in these expressions.

Maximizing the objective, computed using (10), is equivalent to maximizing its logarithm. We reformulate the objective as

$$\ln J = \sum_{k=0}^{K-1} \ln \mu((t_k, t_{k+1}), \rho) + \sum_{k=0}^{K} \ln \mu(\{t_k\}, \rho) \tag{23}$$

after substitution using (21) and (22) for practicality. Furthermore, a final simplification of (23) is performed using (7) to write

$$\ln J = f_0 \sum_{k=0}^{K-1} \int_{t_k}^{t_{k+1}} \ln \rho(t) dt + \sum_{k=0}^{K} \ln \rho(t_k), \tag{24}$$

where $f_0$ is the sampling frequency of the measurements.

The governing optimization problem is formed by maximizing (24) subject to constraints which enforce the state and measurement models in (16) and (17). This is equivalent to the problem

$$
\begin{aligned}
\min_{x,v,w}. \quad & f_0 \sum_{k=0}^{K-1} \int_{t_k}^{t_{k+1}} \frac{v(t)^2}{2\sigma_p^2} dt + \sum_{k=0}^{K} \left( \frac{v(t_k)^2}{2\sigma_p^2} + \frac{w(t_k)^2}{2\sigma_m^2} \right) \\
\text{s.t.} \quad & \dot{x}(t) = \begin{pmatrix} 0 & 1 \\ 0 & 0 \end{pmatrix} x(t) + \begin{pmatrix} 0 \\ 1 \end{pmatrix} v(t) \quad \text{for } t \in \mathscr{T} \\
& y(t) = \begin{pmatrix} 1 & 0 \end{pmatrix} x(t) + w(t) \quad \text{for } t \in \mathscr{T}_M,
\end{aligned}
\tag{25}
$$

in this simple example. We postpone the analysis of this problem, instead solving a more general problem in the next section first, and return to this while reviewing this in the Illustrative Examples. However, we preview the resulting spline which arises from its solution to build intuition before moving forward. Interestingly, the optimal solution in this system takes the form of a cubic spline. We demonstrate the optimal spline behavior on simulated data in the following subsection, and discuss additional simulations of this system in the following subsection.

## Review and preview—Cubic splines

To review briefly, a cubic spline is a function $\hat{r}(t)$ that is represented by the expression

$$
\hat{r}(t) = a(t - t_k)^3 + b(t - t_k)^2 + c(t - t_k) + d
\tag{26}
$$

on the interval $[t_k, t_{k+1}]$ where $a$, $b$, $c$, and $d$ are constants. The constants defining the spline on a given interval are allowed to differ from interval-to-interval. In the case of a NCS as discussed prior, the constants are selected along the intervals to fit the data while maintaining continuity in the function and its derivatives. Other constraints can be enforced in lieu of those that occur in the construction of natural splines, and indeed in our new method alternative requirements are automatically produced in the optimization process instead.

We simulate the point mass dynamics with noisy measurements. We set $\sigma_p = 4$ and $\sigma_m = 1$ for the purposes of this demonstration. The data consists of a simulated trajectory, constructed using discrete updates such that

$$
r(t + dt) = r(t) + \dot{r}(t)dt + \frac{1}{2} v(t)dt^2,
\tag{27}
$$

and

$$
\dot{r}(t + dt) = \dot{r}(t) + v(t)dt.
\tag{28}
$$

For each simulation time $t$, the simulated acceleration $v(t)$ is randomly drawn from a zero-mean Gaussian distribution with variance $\sigma_p^2$, and $dt$ was set to 0.01 time units.

The trajectory is generated using a starting time $t_0 = 0$ and a final time $t_K = 10$, with $r(0) = 10$ and $\dot{r}(0) = 0$ used to initialize the trajectory. Simulated measurements are collected with sampling frequency $f_0 = 5 \frac{\text{samples}}{\text{time unit}}$ using the relation

$$
y(t_k) = r(t_k) + w(t_k).
\tag{29}
$$

The measurement noise $w(t_k)$ is randomly drawn from a zero-mean Gaussian distribution with variance $\sigma_m^2$ for each measurement.

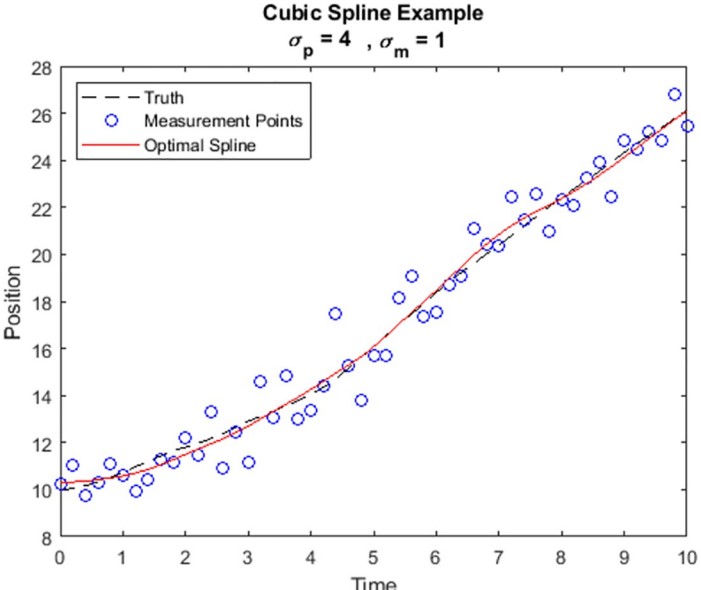

**Fig 1. A simulated trajectory with noisy measurements and an optimal cubic spline solution.**

The simulated noisy measurements are used to construct an optimal spline by solving the optimization problem (24). This spline is shown to be a cubic spline in the solution process which is described later in the illustrative examples. The noiseless trajectory, the simulated measurement points, and the optimal cubic spline are all shown in Fig 1. Notice that the optimization does not force the spline to pass through each of the measurement points. In a simple NCS the spline is required to coincide with measurements at sampling times making it sensitive to noise. In the presence of such noise this can induce oscillations which are not present in the underlying true trajectory, causing accuracy breakdown. On the other hand, the present optimization theory provides both the cubic structure without prior assumption and stitching conditions which help mitigate noise effects from each interval to the next automatically. The derivation of the cubic structure and optimal stitching conditions are developed in detail in the simple particle motion illustrative example that follows later in this work. This preview demonstrates that the result produces a spline that remains close to the true trajectory despite the measurement noise.

A numerical Monte Carlo study was used to evaluate the performance benefit of the optimization spline versus an NCS in circumstances favorable for NCS applications. Even in the case when an NCS is an applicable tool (low measurement noise), the optimization spline over 1000 simulated trajectories improved RMS accuracy by over 37% on average. Further detail on the random simulation is included following the first Illustrative Example when we revisit this motivating problem after developing the general treatment.

## General problem and analysis

In the general treatment we continue to restrict our attention to $\mathscr{T}$ and $\mathscr{T}_M$ as in the motivational example. That is to say, we assume measurements are taken at discrete times with some sampling frequency $f_0$ and we focus on enriching the data between the first and final measurements, inclusively. However, we modify the concepts utilized to formulate (25) to allow general

stochastic distributions, dynamics, and measurements. The resulting problem is written as

$$\max_{x,v,w} \quad f_0 \sum_{k=0}^{K-1} \int_{t_k}^{t_{k+1}} \ln\rho(t)dt + \sum_{k=0}^{K} \ln\rho(t_k)$$

$$\text{s.t.} \quad \dot{x}(t) = f(t,x(t)) + v(t,v(t)) \text{ for } t \in \mathscr{T} \tag{30}$$

$$y(t) = g(t,\dot{x}(t)) + h(t,x(t)) + \xi(t,w(t)) \text{ for } t \in \mathscr{T}_M.$$

Analysis of the problem proceeds in two steps. First, we transform (30) into an unconstrained optimization problem through use of the Langrangian and dual variables. Second, we apply techniques from the Calculus of Variations (CoV) to derive necessary conditions on optimal solutions. Applying the CoV produces two types of optimal conditions: ordinary differential equations that must be satisfied on $\mathscr{T} - \mathscr{T}_M$, and algebraic relations which must be satisfied on $\mathscr{T}_M$. The solution to the differential equations produces the spline structure automatically, and the algebraic relations provide boundary conditions to constrain the general solutions and in doing so fit the spline to a specific data set.

It is useful to introduce compressed notation for clarity in development of the Lagrangian. We define

$$\phi(t) = \dot{x}(t) - f(t,x(t)) - v(t,v(t)), \tag{31}$$

and

$$\psi(t) = y(t) - g(t,\dot{x}(t)) - h(t,x(t)) - \xi(t,w(t)). \tag{32}$$

We introduce two sets of dual variables $\lambda$ and $\eta$. The first, $\lambda$, is a function of time defined on $\mathscr{T}$ and relates to the first set of constraints involving the dynamics (31). The second, $\eta$, is a discrete function of time defined only for times in $\mathscr{T}_M$ and corresponds to the second set of constraints relating to the measurements (32). Problem (30) is reformulated using the Lagrangian objective

$$L = \quad f_0 \sum_{k=0}^{K-1} \int_{t_k}^{t_{k+1}} \ln\rho(t) - \lambda(t)^T \phi(t)dt$$

$$+ \sum_{k=0}^{K} \ln\rho(t_k) - \lambda(t_k)^T \phi(t_k) - \eta(t_k)^T \psi(t_k), \tag{33}$$

which yields the equivalent unconstrained optimization

$$\max_{x,v,w,\lambda,\eta} L. \tag{34}$$

The optimal conditions that are necessary for a valid solution contain many Jacobian matrices and one-sided limits. We define notation conventions that allow us to express these concepts efficiently. If $f$ is a differentiable vector-valued function from $\mathbb{R}^n$ to $\mathbb{R}^m$, then we write

$$f(u+\delta u) = f(u) + \frac{\partial f}{\partial u}(u)\delta u + \text{h.o.t.} \tag{35}$$

In this expression $\frac{\partial f}{\partial u}$ is a linear operator of dimension $m \times n$ and is the Jacobian of $f$ with respect to $u$. If $x$ is a function of time, then we define for $\varepsilon > 0$

$$x(t^-) = \lim_{\varepsilon \to 0} x(t-\varepsilon), \quad x(t^+) = \lim_{\varepsilon \to 0} x(t+\varepsilon) \tag{36}$$

as one-sided limits.

The optimization conditions which characterize solutions of (34) are stated in Theorem 1.

**Theorem 1** *Assume x is continuous for all $t \in \mathcal{T}$. If x, v, w, λ, and η are an optimal solution of* (34), *then the following must hold*:

$$\frac{\partial \ln \rho_v}{\partial v} + \frac{\partial v^T}{\partial v} \lambda(t) = 0 \tag{37}$$

$$\dot{\lambda}(t) + \frac{\partial f^T}{\partial x} \lambda(t) = 0 \tag{38}$$

$$\dot{x}(t) - f(t, x(t)) - v(t, \nu(t)) = 0, \tag{39}$$

for all $t \in \mathcal{T} - \mathcal{T}_M$, and

$$\frac{\partial \ln \rho_v}{\partial v} + \frac{\partial v^T}{\partial v} \lambda(t_k) = 0 \tag{40}$$

$$\frac{\partial \ln \rho_w}{\partial w} + \frac{\partial \xi^T}{\partial w} \eta(t_k) = 0 \tag{41}$$

$$\lambda(t_k) - \frac{\partial g^T}{\partial \dot{x}} \eta(t_k) = 0 \tag{42}$$

$$\frac{\partial f^T}{\partial x} \lambda(t_k) + \frac{\partial h^T}{\partial x} \eta(t_k) + f_0 \lambda(t_k^+) - f_0 \lambda(t_k^-) = 0 \tag{43}$$

$$y(t_k) - g(t_k, \dot{x}(t_k)) - h(t_k, x(t_k)) - \xi(t_k, w(t_k)) = 0 \tag{44}$$

$$\dot{x}(t_k) - f(t_k, x(t_k)) - v(t_k, \nu(t_k)) = 0, \tag{45}$$

for all $t_k \in \mathcal{T}_M$ (When $\mathcal{T} = [t_0, t_K]$, then define $\lambda(t_0^-) = \lambda(t_K^+) = 0$ for simplicity).

The proof of Theorem 1 is provided by standard application of the CoV. This generates a natural derivation of the full set of optimization conditions listed within the theorem. An outline of the proof is provided in the S1 Appendix; for technical details of the CoV the reader is referred to [15].

Eqs (37)–(39) define the differential equations that are satisfied on each interval between measurements. This system of equations depends only on x, λ, and v. Therefore we conclude that the general structure of the optimal spline is independent of the measurements, and it only depends on the dynamical system and the nature of the corresponding stochastic process. This is an interesting result, establishing that a system of stochastic dynamics comes equipped with an optimal spline structure, prior to the collection of measurements.

Eqs (40)–(45) describe the discrete conditions that must hold from one measurement-free interval to the next. This system of algebraic equations provides the constraints for the general solutions to the differential equations, and its solution ultimately enriches the specific set of data by optimally fitting the spline to the measurements.

Constraint (43) provides one-sided limits of λ at each measurement point, and these values are used as boundary conditions to fit solutions of (38). The dual variable λ need not be continuous at the measurement points in general, and therefore the right and left handed limits are critical. On the other hand, under the assumption that x is continuous at each of the

measurement points, we have

$$x(t_k^{\pm}) = x(t_k) \tag{46}$$

for all $t_k \in \mathscr{T}_M$. The values $\{x(t_k)\}_{k=0}^K$ are solved for using the algebraic constraints, and these values provide the boundary conditions for (39). The assumption that $x$ is continuous is a natural, physically motivated one; if $f$ and $v$ are bounded then $x$ is necessarily continuous given (1).

## Illustrative examples

We provide illustrative examples, where the first two are chosen as specific instances of the third. We focus our attention in the first three examples on linear dynamical systems with Gaussian stochastic processes. These are often good approximations of real systems, and are the standards assumptions under which Kalman filters/smoothers are derived. These examples demonstrate that our theory produces optimal splines that are automatically adapted to the dynamics of an underlying system.

For the first case, we return to the analysis of the motivational example from earlier. We show that the optimal spline is cubic, develop the fitting conditions, and exhibit its behavior on various simulated noisy measurements. For the second case, we complicate the dynamics by considering a harmonic oscillator and repeat the analysis. In this example, we show explicitly that the structure of the optimal spline radically changes with the dynamics. The piecewise structure of the spline becomes a type of modified harmonic, and is no longer cubic. Finally, for the third case we consider the Kalman smoothing problem for general linear systems. We compute the specific form of Theorem 1 and show that the linear form of the filter, which is an explicit assumption in Kalman filtering theory, is merely a natural consequence of our developed framework.

We complete the illustrative examples by considering two additional scenarios in which the optimal filter is non-linear. The first of these problems considers a case with a non-Gaussian process, and the second includes a problem with non-linear dynamics. In these examples the linear Kalman approach is no longer optimal and indeed a non-linear solution arises automatically from the new framework.

## Simple particle motion

We restate the dynamics (16), measurements (17), and stochastic distributions (21)–(22) here for convenience:

$$\dot{x}(t) = \begin{pmatrix} 0 & 1 \\ 0 & 0 \end{pmatrix} x(t) + \begin{pmatrix} 0 \\ 1 \end{pmatrix} v(t),$$

$$y(t) = \begin{pmatrix} 1 & 0 \end{pmatrix} x(t) + w(t),$$

$$\rho_v(t, v(t)) = \frac{1}{\sqrt{2\pi}\sigma_p} e^{-\frac{1}{2\sigma_p^2} v(t)^2},$$

and

$$\rho_w(t, w(t)) = \frac{1}{\sqrt{2\pi}\sigma_m} e^{-\frac{1}{2\sigma_m^2} w(t)^2}.$$

We apply Theorem 1 to derive the optimal splines and stitching conditions. We begin with the differential equations describing the optimal spline structure in (37)–(39):

$$-\frac{1}{\sigma_p^2}v(t) + (\,0 \quad 1\,)\lambda(t) = 0 \tag{47}$$

$$\dot{\lambda}(t) + \begin{pmatrix} 0 & 0 \\ 1 & 0 \end{pmatrix}\lambda(t) = 0 \tag{48}$$

$$\dot{x}(t) - \begin{pmatrix} 0 & 1 \\ 0 & 0 \end{pmatrix}x(t) - \begin{pmatrix} 0 \\ 1 \end{pmatrix}v(t) = 0. \tag{49}$$

This must hold for all $t$ in intervals of the form $(t_k, t_{k+1})$. The solution on the interval $(t_k, t_{k+1})$ is of the general form

$$v(t) = a_k(t - t_k) + b_k, \tag{50}$$

$$\lambda(t) = \begin{pmatrix} a_k \\ a_k(t - t_k) + b_k \end{pmatrix}, \tag{51}$$

and

$$x(t) = \begin{pmatrix} \dfrac{a_k\sigma_p^2}{6}(t - t_k)^3 + \dfrac{b_k\sigma_p^2}{2}(t - t_k)^2 + c_k(t - t_k) + d_k \\[2ex] \dfrac{a_k\sigma_p^2}{2}(t - t_k)^2 + b_k\sigma_p^2(t - t_k) + c_k \end{pmatrix}, \tag{52}$$

where $a_k$, $b_k$, $c_k$, and $d_k$ are a set of arbitrary constants. Notice that the particle position, the first component of $x(t)$, has a cubic form that is consistent with (26). This establishes explicitly how cubic splines arise naturally in the stochastic system defined by (16) and (21) in this framework. As an added benefit, the second component of $x(t)$ provides an optimal estimate of the particle velocity.

The algebraic conditions in (40)–(45) are written as

$$-\frac{1}{\sigma_p^2}v(t_k) + (\,0 \quad 1\,)\lambda(t_k) = 0 \tag{53}$$

$$-\frac{1}{\sigma_m^2}w(t_k) + \eta(t_k) = 0 \tag{54}$$

$$\lambda(t_k) = 0 \tag{55}$$

$$\begin{pmatrix} 0 & 0 \\ 1 & 0 \end{pmatrix}\lambda(t_k) + \begin{pmatrix} 1 \\ 0 \end{pmatrix}\eta(t_k) + f_0\lambda(t_k^+) - f_0\lambda(t_k^-) = 0 \tag{56}$$

$$y(t_k) - (\,1 \quad 0\,)x(t_k) - w(t_k) = 0 \tag{57}$$

$$\dot{x}(t_k) - \begin{pmatrix} 0 & 1 \\ 0 & 0 \end{pmatrix}x(t_k) - \begin{pmatrix} 0 \\ 1 \end{pmatrix}v(t_k) = 0, \tag{58}$$

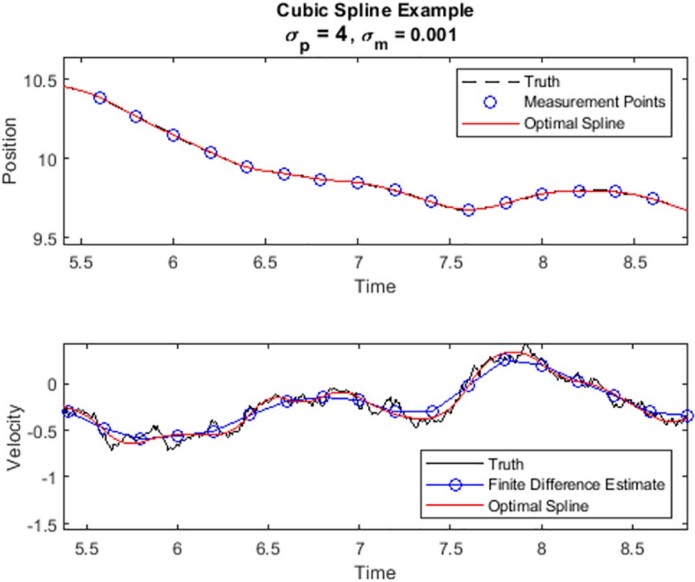

**Fig 2. Simulated point mass trajectory with high accuracy measurements.** [Top] Particle Position. [Bottom] Particle Velocity.

for each $t_k \in \mathscr{T}_M$. This set of algebraic equations is used to constrain the general constants on the intervals to construct the spline which best enriches the measurements. A desirable attribute of splines is that they provide *infinite resolution* while only requiring *finite representations*. Our optimal spline produces continuous time estimates but is simply represented by a finite set of $4K$ constants.

In Fig 2 we show a simulated set of high accuracy measurements where $\sigma_m \ll \sigma_p$. The simulation was generated using the same procedure as in the motivational example. The lower portion of Fig 2 depicts the true velocity, the velocity estimate arising from a piece-wise linear fit (finite differences), and the velocity estimate automatically produced by the new optimal spline. In Fig 3 we show results from a simulation with much lower measurement accuracy, i.e., with larger $\sigma_m$ than in Fig 2. Compare the behavior of the optimal spline in this second simulation with that of the first. The optimal spline no longer adheres to each measurement point individually, but it does continue to approximate the true trajectory well.

Importantly, the constants $\{a_k, b_k, c_k, d_k\}_{k=0}^K$ are linearly related to the measured observations. Therefore, the optimally estimated state at any time $t$ for this system is linear with respect to the measurements. This is consistent with Kalman optimality, where it is known that a linear filter is the optimal choice for linear dynamical systems subject to Gaussian processes. In our treatment the linear dependence of the estimated state on the measured values is a natural consequence following from the optimization framework rather than being imposed from the outset. Before considering additional illustrative examples we evaluate the performance of the optimal cubic spline compared to the performance of a basic NCS in Monte Carlo simulations of simple particle motion.

## Simulation—Optimal spline vs NCS

The parameters of the Monte Carlo simulation is first described, and then the results of analyzing the random data set are presented with brief discussion. Ultimately, we observed an

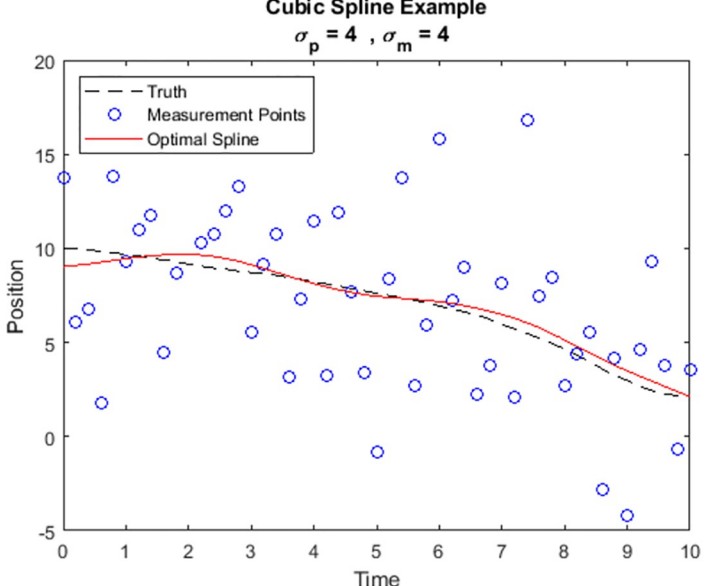

**Fig 3. Simulated point mass trajectory with low accuracy measurements.**

average accuracy improvement of 37.82% in RMS error when using the optimal spline instead of the simple NCS.

**Simulation—Description.** We simulate a point mass with noisy measurements subject to stochastic forcing. We set $\sigma_p = 4.0$ and $\sigma_m = 0.1$ for the purposes of the investigation. Measurement noise levels were specified to be relatively low so that a simple NCS would be an applicable alternative method to use for comparison. The generated data set consists of simulated trajectories, constructed using discrete updates as in the Motivational Example (Eqs (27) and (28)). For each simulation time $t$, $v(t)$ is randomly drawn from a zero-mean Gaussian distribution with variance $\sigma_p^2$. The random trajectories are generated with $dt$ set to 0.0039 seconds which corresponds to a state update frequency $f = 256$ Hz for the purposes of simulation. For each trajectory we set a starting time of $t_0 = 0$ seconds and a terminal time of $t_K = 30$ seconds. The initial position and velocity of the particle in each trajectory are drawn from normal distributions. Explicitly,

$$r(0) \sim N(0, 1) \tag{59}$$

and

$$\dot{r}(0) \sim N(0, 1). \tag{60}$$

Simulated measurements are collected with sampling frequency $f_s = 2$ Hz using the relation

$$y(t_k) = x(t_k) + w(t_k), \tag{61}$$

with

$$t_k = \frac{k}{f_s}. \tag{62}$$

The measurement noise $w(t_k)$ is randomly drawn from a zero-mean Gaussian distribution with variance $\sigma_m^2$ for each measurement. A collection of $N_{\text{trials}} = 1000$ random trajectories were simulated and stored along with their underlying truth (before sampling and noise addition).

For each sampled noisy trajectory the proposed optimization spline and an NCS were fit to the data to estimate the original true, noise-free trajectory at the simulation update rate of 256 Hz. Denoting $\hat{r}(t)$ as an estimate using a spline, the RMS error of the fit was computed using

$$\epsilon^2 = \sum_{n=0}^{N-1} \frac{(\hat{r}(t_n) - r(t_n))^2}{N}, \tag{63}$$

where $r(t_n)$ denotes the true trajectory value at time $t_n$ and $N$ denotes the total number of points in the true trajectory. The $n$-th simulation time $t_n$ is given by

$$t_n = \frac{n}{f}. \tag{64}$$

The percentage improvement is defined as

$$\text{Improvement } \% = 100 \times \left( 1 - \frac{\epsilon_{os}}{\epsilon_{ncs}} \right), \tag{65}$$

where $\epsilon_{os}$ and $\epsilon_{ncs}$ denote the RMS error of the optimization spline and NCS, respectively.

**Simulation—Results.** In Figs 4 and 5 we show example results from two of the simulated trajectories randomly selected from the data set. We have simulated minor measurement noise and therefore a nuanced comparison is difficult to make while examining the trajectories and splines. Therefore, we have included a second subplot in each case to show the error of each spline over the course of the trajectory and this shows how, even in the presence of minor measurement noise where NCS is most applicable, the optimization spline produces less volatility and more consistent performance.

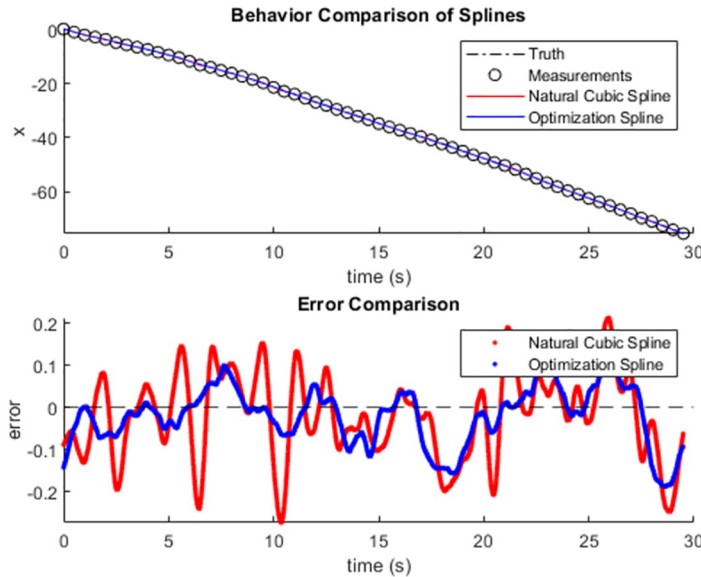

**Fig 4. Trajectory number 100 from the simulated data set for comparison.** The true trajectory, measured data, the optimization spline, and NCS (top), and the error over time of each spline (bottom) are shown.

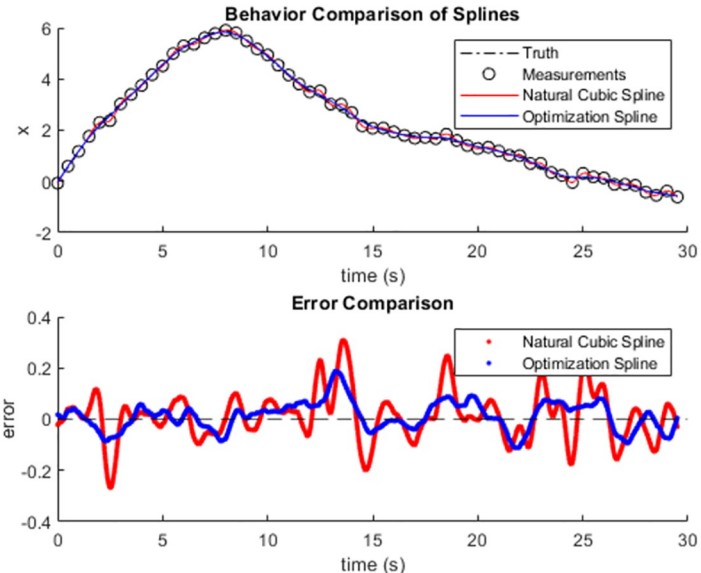

**Fig 5. Trajectory number 530 from the simulated data set for comparison.** The true trajectory, measured data, the optimization spline, and NCS (top), and the error over time of each spline (bottom) are shown.

In Fig 6 we provide the histogram of performance improvements over the set of stochastic trajectories. In all simulated cases the optimization spline outperformed the NCS, and on average provided a 37.82% reduction in error when compared to the NCS. The histogram appears to be loosely Gaussian, with a minimal realized improvement of 15.38% and a maximum realized improvement of 60.21%.

**Simulation—Concluding notes.**   The optimization based splines described in this work outperformed the natural cubic spline in all 1000 trials simulated. This is appealing as the

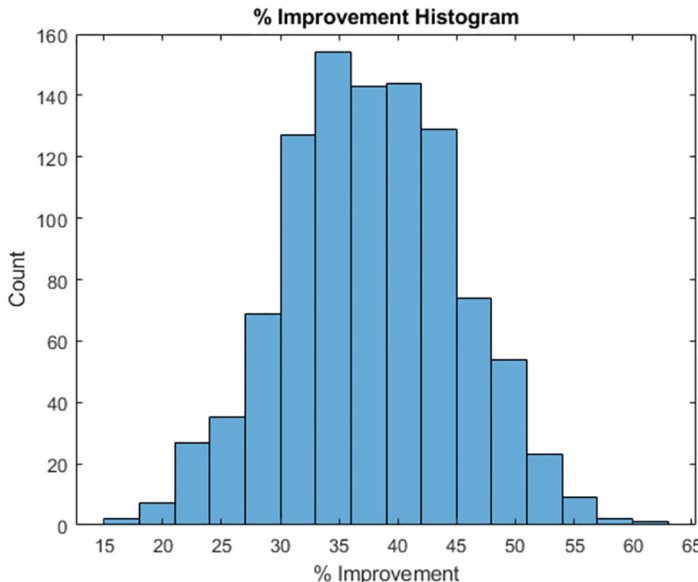

**Fig 6. Histogram of improvement % over simulated random trajectories.**

optimization based splines adapt based on the underlying dynamics and statistics of the process and measurement noises whereas the NCS does not. The simulation modeled low measurement noise trajectories. In the presence of increased measurement noise the NCS performance would suffer and the optimization splines would out-perform the NCS-based ones by even greater margins as the noise levels increased. In high noise scenarios NCS becomes less applicable and alternative data enrichment techniques, such as Kalman smoothing or GPR, should be used instead if further simulated comparison studies were conducted.

### Harmonic oscillators

We repeat the example in the previous section with the minor modification of replacing the governing dynamics by a harmonic oscillator with natural frequency $\omega^2$. The modified system is written as

$$\dot{x}(t) = \begin{pmatrix} 0 & 1 \\ -\omega^2 & 0 \end{pmatrix} x(t) + \begin{pmatrix} 0 \\ 1 \end{pmatrix} v(t), \tag{66}$$

$$y(t) = \begin{pmatrix} 1 & 0 \end{pmatrix} x(t) + w(t), \tag{67}$$

$$\rho_v(t, v(t)) = \frac{1}{\sqrt{2\pi}\sigma_p} e^{-\frac{1}{2\sigma_p^2}v(t)^2}, \tag{68}$$

and

$$\rho_w(t, w(t)) = \frac{1}{\sqrt{2\pi}\sigma_m} e^{-\frac{1}{2\sigma_m^2}w(t)^2}. \tag{69}$$

Again we apply Theorem 1. First, examining the differential equations, we consider

$$-\frac{1}{\sigma_p^2} v(t) + \begin{pmatrix} 0 & 1 \end{pmatrix} \lambda(t) = 0 \tag{70}$$

$$\dot{\lambda}(t) + \begin{pmatrix} 0 & -\omega^2 \\ 1 & 0 \end{pmatrix} \lambda(t) = 0 \tag{71}$$

$$\dot{x}(t) - \begin{pmatrix} 0 & 1 \\ -\omega^2 & 0 \end{pmatrix} x(t) - \begin{pmatrix} 0 \\ 1 \end{pmatrix} v(t) = 0. \tag{72}$$

The solution on the interval $(t_k, t_{k+1})$ is given by

$$v(t) = a_k \sin \omega t + b_k \cos \omega t, \tag{73}$$

$$\lambda(t) = a_k \begin{pmatrix} -\omega \cos \omega t \\ \sin \omega t \end{pmatrix} + b_k \begin{pmatrix} \omega \sin \omega t \\ \cos \omega t \end{pmatrix}, \tag{74}$$

and

$$x(t) = \begin{pmatrix} \left(\frac{\sigma_p^2 b_k}{2\omega} t + c_k\right) \sin \omega t + \left(-\frac{\sigma_p^2 a_k}{2\omega} t + d_k\right) \cos \omega t \\ \left(\frac{\sigma_p^2 a_k}{2} t + \frac{\sigma_p^2 b_k}{2\omega} - \omega d_k\right) \sin \omega t + \left(\frac{\sigma_p^2 b_k}{2} t - \frac{\sigma_p^2 a_k}{2\omega} + \omega c_k\right) \cos \omega t \end{pmatrix} \tag{75}$$

where $a_k$, $b_k$, $c_k$, and $d_k$ are again arbitrary constants arising in the general solution. It is interesting to note the different structure of this solution compared to the previous example, resulting from a small modification of the dynamics. In particular, this optimal spline is a piece-wise function of the form

$$\left(\frac{\sigma_p^2 b_k}{2\omega}t + c_k\right)\sin \omega t + \left(-\frac{\sigma_p^2 a_k}{2\omega}t + d_k\right)\cos \omega t, \tag{76}$$

which we refer to as a *modified harmonic.*

The algebraic conditions for this system are written as

$$-\frac{1}{\sigma_p^2}v(t_k) + (\begin{matrix}0 & 1\end{matrix})\lambda(t_k) = 0 \tag{77}$$

$$-\frac{1}{\sigma_m^2}w(t_k) + \eta(t_k) = 0 \tag{78}$$

$$\lambda(t_k) = 0 \tag{79}$$

$$\begin{pmatrix}0 & -\omega^2 \\ 1 & 0\end{pmatrix}\lambda(t_k) + \begin{pmatrix}1 \\ 0\end{pmatrix}\eta(t_k) + f_0\lambda(t_k^+) - f_0\lambda(t_k^-) = 0 \tag{80}$$

$$y(t_k) - (\begin{matrix}1 & 0\end{matrix})x(t_k) - w(t_k) = 0 \tag{81}$$

$$\dot{x}(t_k) - \begin{pmatrix}0 & 1 \\ -\omega^2 & 0\end{pmatrix}x(t_k) - \begin{pmatrix}0 \\ 1\end{pmatrix}v(t_k) = 0, \tag{82}$$

for each $t_k \in \mathcal{T}_M$. These equations are used to constrain the general solution and construct a spline for the specific data set. In this example we still have a system of linear dynamics and measurements with Gaussian stochastic processes, and thus the new framework again derives a linear relationship between the measured data and optimal spline. We simulate a stochastic harmonic oscillator and perform a representative data enrichment in Fig 7.

We have emphasized the adaptive nature of the splines in the new framework. It is interesting to make a direct comparison of the matched spline to that of the cubic spline arising from the simple particle dynamics. In Fig 8 we repeat the simulation of the stochastic harmonic oscillator and fit the data using an optimal cubic spline and an optimal modified harmonic spline.

In both examples covered thus far we observed that the spline was linearly dependent on the measurements. In the next section we analyze an example with general linear dynamics and measurements subject to Gaussian additive noise. These are the class of systems for which the Kalman filter was derived and is known to be optimal. We include this illustrative example to provide new perspectives on smoothing. In particular, we demonstrate that our framework provides an alternative means of smoothing data that circumvents the cumbersome matrix-valued and nonlinear (differential Riccati) equations that arise in Kalman smoothing. Indeed, for linear systems Theorem 1 yields linear differential and linear algebraic equations which, once solved, result in linear dependence of the splines on measurement data.

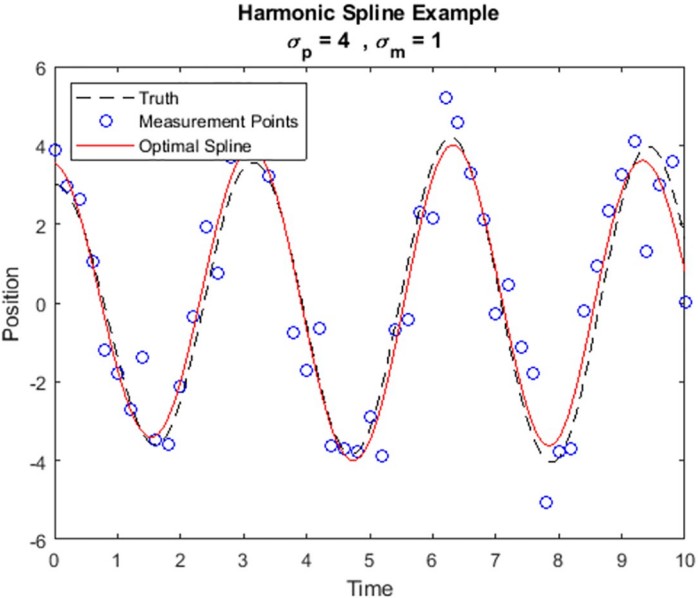

**Fig 7. Simulated stochastic harmonic oscillator trajectory with optimal spline.**

## General linear systems

We consider a linear system

$$\dot{x}(t) = Ax(t) + Bv(t), \tag{83}$$

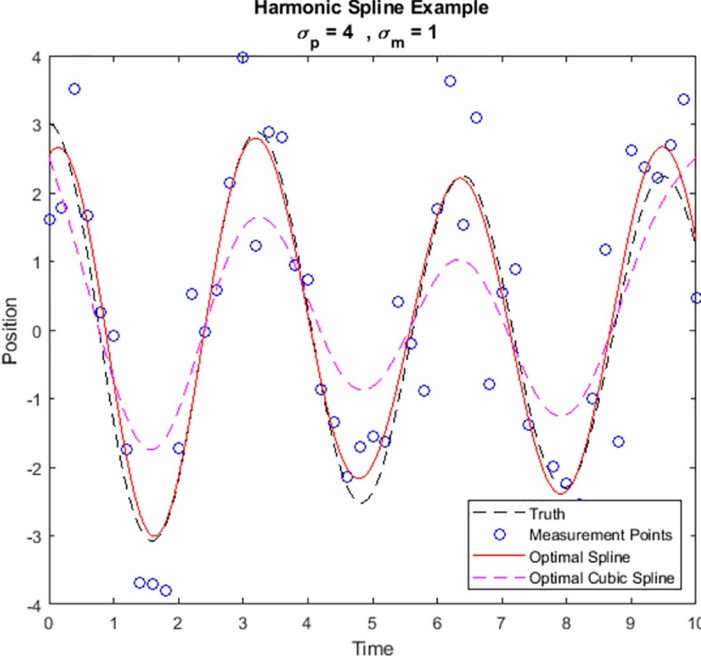

**Fig 8. Simulated stochastic harmonic oscillator trajectory with both harmonic and cubic optimal splines for comparison.**

$$y(t) = Cx(t) + Dw(t),\tag{84}$$

and Gaussian random processes

$$\rho_v(t, v(t)) = \frac{1}{\sqrt{2\pi}^{n_v} \det(Q)^{\frac{1}{2}}} e^{-\frac{1}{2}v(t)^T Q^{-1} v(t)}\tag{85}$$

$$\rho_w(t, w(t)) = \frac{1}{\sqrt{2\pi}^{n_w} \det(R)^{\frac{1}{2}}} e^{-\frac{1}{2}w(t)^T R^{-1} w(t)}.\tag{86}$$

In (85), we take $Q$ to be the covariance of $v(t)$. Similarly in (86) $R$ denotes the covariance of $w(t)$. The matrix $A$ is of dimension $n_x \times n_x$, $B$ is of dimension $n_x \times n_v$, $C$ is of dimension $n_y \times n_x$, and $D$ is of dimension $n_y \times n_w$.

As we did in the previous two examples, we apply Theorem 1 and consider the differential equations that govern the spline,

$$-Q^{-1}v(t) + B^T\lambda(t) = 0\tag{87}$$

$$\dot{\lambda}(t) + A^T\lambda(t) = 0\tag{88}$$

$$\dot{x}(t) - Ax(t) - Bv(t) = 0.\tag{89}$$

This system is solved on $(t_k, t_{k+1})$ by

$$v(t) = QB^T e^{-A^T(t-t_k)} c_k^{(\lambda)},\tag{90}$$

$$\lambda(t) = e^{-A^T(t-t_k)} c_k^{(\lambda)},\tag{91}$$

and

$$x(t) = e^{A(t-t_k)}\left[\int_0^{t-t_k} e^{-As} BQB^T e^{-A^T s} ds\, c_k^{(\lambda)} + c_k^{(x)}\right],\tag{92}$$

where $c_k^{(\lambda)}$ and $c_k^{(x)}$ are constant vectors. This solution describes the structure of the optimal spline for a general linear system with Gaussian processes.

Constraining the constant coefficients of the spline requires use of the algebraic equations as we have shown in previous examples, and these are written as

$$-Q^{-1}v(t_k) + B^T\lambda(t_k) = 0\tag{93}$$

$$-R^{-1}w(t_k) + D^T\eta(t_k) = 0\tag{94}$$

$$\lambda(t_k) = 0\tag{95}$$

$$A^T\lambda(t_k) + C^T\eta(t_k) + f_0\lambda(t_k^+) - f_0\lambda(t_k^-) = 0\tag{96}$$

$$y(t_k) - Cx(t_k) - Dw(t_k) = 0\tag{97}$$

$$\dot{x}(t_k) - Ax(t_k) - Bv(t_k) = 0,\tag{98}$$

for each $t_k \in \mathscr{T}_M$. Therefore construction of the optimal spline requires only the solution of a *linear* system in this treatment. This is not only numerically appealing but we reiterate that it results in closed-form solutions that provide infinite time resolution through a representation as a finite set of constants.

### An example with non-Gaussian process noise

We move away from Kalman-type examples to further demonstrate benefits of the new framework. We revisit the simple particle dynamics under forcing by a non-Gaussian stochastic process. A linear filter is non-optimal in this scenario, and indeed the new framework produces a non-linear optimal solution automatically. The modified simple particle system is described by

$$\dot{x}(t) = \begin{pmatrix} 0 & 1 \\ 0 & 0 \end{pmatrix} x(t) + \begin{pmatrix} 0 \\ 1 \end{pmatrix} v(t), \tag{99}$$

$$y(t) = \begin{pmatrix} 1 & 0 \end{pmatrix} x(t) + w(t), \tag{100}$$

$$\rho_v(t, v(t)) = c_v e^{-\frac{1}{2}\left(\frac{v(t)}{\sigma_p}\right)^{2\alpha}}, \tag{101}$$

and

$$\rho_w(t, w(t)) = \frac{1}{\sqrt{2\pi}\sigma_m} e^{-\frac{1}{2\sigma_m^2}w(t)^2}. \tag{102}$$

The modification introduces a parameter $\alpha$ in the exponent of $\rho_v$, and it is assumed to be a positive integer greater than 1. The leading constant $c_\alpha$ normalizes the distribution and is a function of the parameter, but its specific value is immaterial in our discussion.

The differential equations arising from Theorem 1 are written as

$$-\frac{\alpha}{\sigma_p^{2\alpha}} v(t)^{2\alpha-1} + \begin{pmatrix} 0 & 1 \end{pmatrix} \lambda(t) = 0 \tag{103}$$

$$\dot{\lambda}(t) + \begin{pmatrix} 0 & 0 \\ 1 & 0 \end{pmatrix} \lambda(t) = 0 \tag{104}$$

$$\dot{x}(t) - \begin{pmatrix} 0 & 1 \\ 0 & 0 \end{pmatrix} x(t) - \begin{pmatrix} 0 \\ 1 \end{pmatrix} v(t) = 0. \tag{105}$$

When $\alpha = 1$ we recover the Gaussian result. For $\alpha > 1$ the system is non-linear but admits a closed form solution. Eq (104) is solved by

$$\lambda(t) = \begin{pmatrix} a \\ at + b \end{pmatrix}, \tag{106}$$

where $a$ and $b$ are arbitrary constants. Substitution of $\lambda(t)$ in (103) yields

$$v(t) = \left(\frac{\sigma_p^{2\alpha}}{\alpha}\right)^{\frac{1}{2\alpha-1}} (at + b)^{\frac{1}{2\alpha-1}}. \tag{107}$$

This expression is substituted in (105), yielding

$$
x(t) = \begin{pmatrix} \dfrac{(2\alpha-1)^2}{(4\alpha-1)(2\alpha a^2)} \left(\dfrac{\sigma_p^{2\alpha}}{\alpha}\right)^{\frac{1}{2\alpha-1}} (at+b)^{\frac{4\alpha-1}{2\alpha-1}} + \dfrac{c}{a}t + d \\[3mm] \dfrac{2\alpha-1}{2\alpha a} \left(\dfrac{\sigma_p^{2\alpha}}{\alpha}\right)^{\frac{1}{2\alpha-1}} (at+b)^{\frac{2\alpha}{2\alpha-1}} + c \end{pmatrix}, \tag{108}
$$

where $c$ and $d$ are additional arbitrary constants.

This solution determines the spline structure on each of the measurement-free intervals, and therefore we again solve for $4K$ unknown constants. These are computed with the algebraic equations from Theorem 1, written as

$$
-\frac{\alpha}{\sigma_p^{2\alpha}} v(t_k)^{2\alpha-1} + \begin{pmatrix} 0 & 1 \end{pmatrix} \lambda(t_k) = 0 \tag{109}
$$

$$
-\frac{1}{\sigma_m^2} w(t_k) + \eta(t_k) = 0 \tag{110}
$$

$$
\lambda(t_k) = 0 \tag{111}
$$

$$
\begin{pmatrix} 0 & 0 \\ 1 & 0 \end{pmatrix} \lambda(t_k) + \begin{pmatrix} 1 \\ 0 \end{pmatrix} \eta(t_k) + f_0 \lambda(t_k^+) - f_0 \lambda(t_k^-) = 0 \tag{112}
$$

$$
y(t_k) - \begin{pmatrix} 1 & 0 \end{pmatrix} x(t_k) - w(t_k) = 0 \tag{113}
$$

$$
\dot{x}(t_k) - \begin{pmatrix} 0 & 1 \\ 0 & 0 \end{pmatrix} x(t_k) - \begin{pmatrix} 0 \\ 1 \end{pmatrix} v(t_k) = 0, \tag{114}
$$

for each $t_k \in \mathcal{T}_M$. For $\alpha > 1$ the non-linear relationship between the piecewise spline functions and the unknown constants induces a non-linear relationship between the measured data and the optimal estimate when constructing the spline.

## An example with non-linear dynamics

We conclude the examples with a brief examination of a non-linear dynamical system. We consider the non-linearized dynamics of a simple pendulum forced by a Gaussian process. This system is described by

$$
\ddot{\theta}(t) + \sin\theta = v(t), \tag{115}
$$

where $\theta$ is the angle between the pendulum and direction of the gravitational force. We rewrite

the system in standard form as

$$\dot{x} = \begin{pmatrix} x_2 \\ -\sin(x_1) \end{pmatrix} + \begin{pmatrix} 0 \\ 1 \end{pmatrix} v \tag{116}$$

$$y = \begin{pmatrix} 1 & 0 \end{pmatrix} x + w \tag{117}$$

$$\rho_v(t, v(t)) = \frac{1}{\sqrt{2\pi}\sigma_p} e^{-\frac{1}{2\sigma_p^2} v(t)^2} \tag{118}$$

$$\rho_w(t, w(t)) = \frac{1}{\sqrt{2\pi}\sigma_m} e^{-\frac{1}{2\sigma_m^2} w(t)^2}, \tag{119}$$

where

$$x = \begin{pmatrix} \theta \\ \dot{\theta} \end{pmatrix} \tag{120}$$

and $y(t)$ models a noisy measurement of $\theta(t)$.

We will compute the optimal spline equations induced by this system, but we do not solve them in the present work as we are not aware of a closed form solution. However, we emphasize that our spline equations facilitate the application of numerical methods to compute the spline. In short, we demonstrate that our framework reduces the enrichment problem to one which simply requires the solution of a system of non-linear differential equations with boundary constraints provided by the algebraic equations.

We note that

$$f(x) = \begin{pmatrix} x_2 \\ -\sin(x_1) \end{pmatrix}, \tag{121}$$

and

$$\frac{\partial f}{\partial x} = \begin{pmatrix} 0 & 1 \\ -\cos(x_1) & 0 \end{pmatrix}, \tag{122}$$

and we use these expressions to simplify notation in the optimization conditions. The differential equations provided by Theorem 1 are written as

$$-\frac{1}{\sigma_p^2} v(t) + \begin{pmatrix} 0 & 1 \end{pmatrix} \lambda(t) = 0 \tag{123}$$

$$\dot{\lambda}(t) + \frac{\partial f}{\partial x}^T \lambda(t) = 0 \tag{124}$$

$$\dot{x}(t) - f(x(t)) - \begin{pmatrix} 0 \\ 1 \end{pmatrix} v(t) = 0. \tag{125}$$

This is a first order system of non-linear differential equations which describes the general form of the optimal spline. The algebraic equations constraining the general solution are

written as

$$-\frac{1}{\sigma_p^2} v(t_k) + (\begin{array}{cc} 0 & 1 \end{array}) \lambda(t_k) = 0 \tag{126}$$

$$-\frac{1}{\sigma_m^2} w(t_k) + \eta(t_k) = 0 \tag{127}$$

$$\lambda(t_k) = 0 \tag{128}$$

$$\frac{\partial f}{\partial x}^T \lambda(t_k) + \begin{pmatrix} 1 \\ 0 \end{pmatrix} \eta(t_k) + f_0 \lambda(t_k^+) - f_0 \lambda(t_k^-) = 0 \tag{129}$$

$$y(t_k) - (\begin{array}{cc} 1 & 0 \end{array}) x(t_k) - w(t_k) = 0 \tag{130}$$

$$\dot{x}(t_k) - f(x(t_k)) - \begin{pmatrix} 0 \\ 1 \end{pmatrix} v(t_k) = 0 \tag{131}$$

for each $t_k \in \mathcal{T}_M$. At this point the system is fully defined, and one would approach computing its solution using an appropriate numerical technique.

## Conclusions and future work

In this work we have developed a technique for enriching data using dynamical system and measurement models under additive forcing from stochastic processes. We developed an optimization framework that allows the robust modeling of the most general dynamical systems and measurements; one that is not limited to linear dynamics or Gaussian stochastic processes. When restricted to the linear Gaussian case our framework naturally, and without prior assumption, renders a linear mapping between the measurements and the optimal spline. This differs from Kalman filtering/smoothing theory, where the linearity of this mapping is assumed from the outset. With the proposed framework we are able to show that cubic splines are the optimal solution for specific linear systems under the effects of Gaussian processes. These systems were simulated in numerical Monte Carlo experiments to demonstrate that the optimal cubic spline proposed in this work provides on average a 37.82% performance improvement in comparison to the established NCS technique.

The capacity to consider more general classes of dynamical systems and stochastic processes when approaching data enrichment in appealing. It allows for the consideration of a much larger class of systems which may be encountered in real applications without requiring Gaussian or linear approximations. The resulting system of equations in Theorem 1 that governs the optimal enrichment in these circumstances will be non-linear in general, and will produce non-linear data filters. Investigation and development of new non-linear data filters is an interesting topic of future work. Moreover, artificial intelligence (AI) and machine learning (ML) continue to advance rapidly and the need for improved data quality and representations has increased with it. The representation of the optimal splines is intrinsically finite and this may have important implications for feature development to support the training of models that will be built using enriched data through modern AI/ML techniques.

The focus of the present work has been on ordinary temporal stochastic dynamical systems, but we are actively pursuing extending the treatment to spatio-temporal systems and other more general problems. We are deeply interested in applying this treatment to systems

governed by partial, as opposed to ordinary, differential equations. Our initial efforts in achieving this goal have begun to bear fruit and we anticipate that future work will produce high performance techniques for performing data enrichment through multiparameter hyper-surfaces as a generalization of the present one-parameter splines.

## Supporting information

**S1 Appendix. Supporting analysis and proofs.** We include proof of the extension theorem which is used to generate stochastic distributions on continuous time intervals. We review the application of the calculus of variations which is used to derive the requirements included in the main theorem of the present work.
(PDF)

## Author Contributions

**Conceptualization:** Griffin M. Kearney.

**Data curation:** Griffin M. Kearney.

**Formal analysis:** Griffin M. Kearney.

**Investigation:** Griffin M. Kearney.

**Methodology:** Griffin M. Kearney.

**Software:** Griffin M. Kearney.

**Supervision:** Makan Fardad.

**Writing – original draft:** Griffin M. Kearney.

**Writing – review & editing:** Griffin M. Kearney, Makan Fardad.

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
