## [Decision Letter · Decision Letter 0]

25 Jul 2024

PONE-D-24-15724Optimization Based Data Enrichment Using Stochastic Dynamical System ModelsPLOS ONE

Dear Dr. Kearney,

Thank you for submitting your manuscript to PLOS ONE. After careful consideration, we feel that it has merit but does not fully meet PLOS ONE’s publication criteria as it currently stands. Therefore, we invite you to submit a revised version of the manuscript that addresses the points raised during the review process.

**ACADEMIC EDITOR: All the reviewers believe that the paper is well written with publishable contribution. However, a number of minor issues have been identified. Thus, a minor revision is helpful to address all these comments before being published. **

We look forward to receiving your revised manuscript.

Kind regards,

Qichun Zhang, PhD

Academic Editor

PLOS ONE

Reviewers' comments:

Reviewer's Responses to Questions

**Comments to the Author**

1. Is the manuscript technically sound, and do the data support the conclusions?

Reviewer #1: Yes

Reviewer #2: Yes

Reviewer #3: Yes

2. Has the statistical analysis been performed appropriately and rigorously? 

Reviewer #1: Yes

Reviewer #2: Yes

Reviewer #3: Yes

3. Have the authors made all data underlying the findings in their manuscript fully available?

Reviewer #1: Yes

Reviewer #2: Yes

Reviewer #3: Yes

4. Is the manuscript presented in an intelligible fashion and written in standard English?

Reviewer #1: Yes

Reviewer #2: Yes

Reviewer #3: Yes

5. Review Comments to the Author

Reviewer #1: The paper is very well written and has the potential to make a significant contribution to the field.

1. However, the literature review needs to be longer. Enriching the data for system modeling is an important part. Many researchers have discussed this topic.

2. Many readers may not be well-versed in the cubic and cubic spline used in this work, and some reviews need to be fulfilled.

Reviewer #2: The contribution of this manuscript is good. Therefore, my recommendation is to accept (Minor Revision) this manuscript that has Ref. No.: PONE-D-24-15724 after several adjustments must be made before publication.

My specific comments are:

1- In the abstract, the result of this work must be described briefly with data in order to show the effectiveness of the proposed work.

2- The author did not describe the drawbacks of each conventional technique in the introduction paragraph.

3- Please include the references for all equations.

4- In conclusions, please add an enhancement percentage (%) that demonstrates the proposed algorithm efficiency for your method when used with another method.

Reviewer #3: - The paper is technically sound and the methods are clear

- A graphical abstract should be given for the clearer picture of the procedure

- Everything looks clear, sound and understandable

- The appendix is also clear

6. PLOS authors have the option to publish the peer review history of their article (what does this mean?). If published, this will include your full peer review and any attached files.

Reviewer #1: No

Reviewer #2: No

Reviewer #3: No

---

## [Author Response · Author response to Decision Letter 0]

8 Aug 2024

Thank you for the helpful feedback. The comments provided by the reviewers have been addressed in the attached letter, "Response to Reviewers.pdf".

---

## [Decision Letter · Decision Letter 1]

2 Sep 2024

Optimization Based Data Enrichment Using Stochastic Dynamical System Models

PONE-D-24-15724R1

Dear Dr. Kearney,

We’re pleased to inform you that your manuscript has been judged scientifically suitable for publication and will be formally accepted for publication once it meets all outstanding technical requirements.

Kind regards,

Qichun Zhang, PhD

Academic Editor

PLOS ONE

Reviewers' comments:

Reviewer's Responses to Questions

**Comments to the Author**

1. If the authors have adequately addressed your comments raised in a previous round of review and you feel that this manuscript is now acceptable for publication, you may indicate that here to bypass the “Comments to the Author” section, enter your conflict of interest statement in the “Confidential to Editor” section, and submit your "Accept" recommendation.

Reviewer #1: All comments have been addressed

Reviewer #2: All comments have been addressed

Reviewer #3: All comments have been addressed

2. Is the manuscript technically sound, and do the data support the conclusions?

Reviewer #1: Yes

Reviewer #2: Yes

Reviewer #3: Yes

3. Has the statistical analysis been performed appropriately and rigorously? 

Reviewer #1: Yes

Reviewer #2: Yes

Reviewer #3: Yes

4. Have the authors made all data underlying the findings in their manuscript fully available?

Reviewer #1: Yes

Reviewer #2: Yes

Reviewer #3: Yes

5. Is the manuscript presented in an intelligible fashion and written in standard English?

Reviewer #1: Yes

Reviewer #2: Yes

Reviewer #3: Yes

6. Review Comments to the Author

Reviewer #1: The authors improved the paper according to the reviewer's comments. So the article is good enough for publication.

Reviewer #2: The contribution of this revised manuscript is clear and good. All comments from the reviewers were very good, and the author’s answer was very good and satisfied. Therefore, my recommendation is to accept the revised manuscript that has Ref. No.: PONE-D-24-15724R1 for publication.

Reviewer #3: Very well written with all comments addressed. Appreciate the approach the authors have taken to obtain data enrichment.

7. PLOS authors have the option to publish the peer review history of their article (what does this mean?). If published, this will include your full peer review and any attached files.

Reviewer #1: No

Reviewer #2: No

Reviewer #3: No

---

## [Editor Report · Acceptance letter]

10 Sep 2024

PONE-D-24-15724R1 

PLOS ONE

Dear Dr. Kearney, 

I'm pleased to inform you that your manuscript has been deemed suitable for publication in PLOS ONE. Congratulations! Your manuscript is now being handed over to our production team.

Kind regards, 

on behalf of

Prof. Qichun Zhang 

Academic Editor

PLOS ONE